# Morphological and Proteomic Analyses of Soybean Seedling Interaction Mechanism Affected by Fiber Crosslinked with Zinc-Oxide Nanoparticles

**DOI:** 10.3390/ijms23137415

**Published:** 2022-07-03

**Authors:** Setsuko Komatsu, Kazuki Murata, Sayuri Yakeishi, Kazuyuki Shimada, Hisateru Yamaguchi, Keisuke Hitachi, Kunihiro Tsuchida, Rumina Obi, Shoichi Akita, Ryo Fukuda

**Affiliations:** 1Faculty of Environment and Information Sciences, Fukui University of Technology, Fukui 910-8505, Japan; 2R&D Laboratory for Applied Product, Asahi Kasei Corporation, Moriyama 524-0002, Japan; murata.kp@om.asahi-kasei.co.jp (K.M.); yakeishi.sb@om.asahi-kasei.co.jp (S.Y.); shimada.kcf@om.asahi-kasei.co.jp (K.S.); obi.lb@om.asahi-kasei.co.jp (R.O.); akita.sb@om.asahi-kasei.co.jp (S.A.); 3Department of Medical Technology, Yokkaichi Nursing and Medical Care University, Yokkaichi 512-8045, Japan; h-yamaguchi@y-nm.ac.jp; 4Institute for Comprehensive Medical Science, Fujita Health University, Toyoake 470-1192, Japan; hkeisuke@fujita-hu.ac.jp (K.H.); tsuchida@fujita-hu.ac.jp (K.T.); 5Business Promotion Section Business Strategy Department, Bemberg Division, Asahi Kasei Corporation, Osaka 530-8205, Japan; fukuda.rb@om.asahi-kasei.co.jp

**Keywords:** proteomics, soybean, zinc oxide, nanoparticles, fiber

## Abstract

Nanoparticles (NPs) enhance soybean growth; however, their precise mechanism is not clearly understood. To develop a more effective method using NPs for the enhancement of soybean growth, fiber crosslinked with zinc oxide (ZnO) NPs was prepared. The solution of ZnO NPs with 200 nm promoted soybean growth at the concentration of 10 ppm, while fibers crosslinked with ZnO NPs promoted growth at a 1 ppm concentration. Soybeans grown on fiber cross-linked with ZnO NPs had higher Zn content in their roots than those grown in ZnO NPs solution. To study the positive mechanism of fiber crosslinked with ZnO NPs on soybean growth, a proteomic technique was used. Proteins categorized in photosynthesis and secondary metabolism accumulated more in soybeans grown on fiber crosslinked with ZnO NPs than in those grown in ZnO NPs solution. Furthermore, significantly accumulated proteins, which were NADPH oxidoreductase and tubulins, were confirmed using immunoblot analysis. The abundance of NADPH oxidoreductase increased in soybean by ZnO NPs application. These results suggest that fiber crosslinked with ZnO NPs enhances soybean growth through the increase of photosynthesis and secondary metabolism. Additionally, the accumulation of NADPH oxidoreductase might relate to the effect of auxin with fiber crosslinked with ZnO NPs on soybean growth.

## 1. Introduction

Metal oxide nanoparticles (NPs) are added to different products due to their unique properties such as catalytic capacity, carcinogenic activity, and drug delivery capabilities [1]. On the other hand, zinc (Zn) is an essential micronutrient for different kinds of living organisms, including plants. Zn deficiency in crops not only restricts their productivity but also influences human health [2]. Zn oxide (ZnO) NPs are extensively utilized due to their optoelectric characteristics [3]; and in products such as plastics, cement, rubber, batteries, and foods [4]. Extensive utilization of ZnO NPs in different products intensified their discharge into the environment. ZnO NPs are added to cosmetic and sunscreen products due to their efficient ultraviolet absorption and reflective properties [5]. Although the incorporation of NPs into the environment creates serious concerns, ZnO NPs are considered to be safe compared with other kinds of NPs.

ZnO NPs suppressed the growth of *Salicornia persica* [6], *Arabidopsis thaliana* [7], tomato, and wheat [8]. ZnO NPs decreased chlorophyll content, which leads to reduced photosynthesis [9]. Although ZnO NPs caused negative effects on plants, they increased the fresh/dry weight of leaves/roots in coffee [10], the growth/biomass of lettuce [11], the germination/root growth of maize/cabbage [12], as well as the plant growth of wheat [13] and tobacco [14]. These findings indicate that ZnO NPs have both negative and positive effects on plant growth. ZnO NPs and bulk ZnO treatments in a substance type- and method-dependent manner improved soybean growth and yield [15]. Furthermore, ZnO NPs influenced seed yield and antioxidant-defense system in soil-grown soybean [16]. These reports indicated that the application of ZnO NPs improved the growth and productivity of soybean.

The application of ZnO NPs at low concentrations provides a safe low-risk approach to improving growth, physiology, immunity, and productivity in crops. In this study, to develop a more effective method using NPs for the enhancement of soybean growth, fibers crosslinked with a low concentration of ZnO NPs were prepared. Additionally, ZnO NPs mitigated the risk associated with diverse abiotic [17,18,19,20] and biotic [21] stresses. These reports indicated that ZnO NPs might potentially confer stress tolerance in plants. Because soybean is sensitive against flooding stress [22], it was analyzed whether soybean could survive against flooding stress on fiber crosslinked with ZnO NPs in this study. To investigate the outcome of fiber crosslinked with ZnO NPs on soybean growth, morphological changes were measured among different fibers crosslinked with different sizes and concentrations of ZnO NPs. Morphological results were further studied by proteomic analysis to determine its response mechanisms. Moreover, proteomic results were upheld by immunoblot analysis.

## 2. Results

### 2.1. Growth Changes of Soybean Treated by Fibers Crosslinked with ZnO NPs

To develop a more effective method using NPs for the enhancement of soybean growth, three kinds of fibers crosslinked with ZnO NPs were prepared. As fibers, a cupro-long fiber non-woven fabric (BL), a cupro-short fiber spun lace non-woven fabric (BFSL), and a viscose-rayon short fiber spun lace non-woven fabric (RA) were used (Appendix A). Morphological changes of soybean with a variety of fibers crosslinked with ZnO NPs were analyzed to investigate their effects (Appendix A). Silica sand was treated in solution with or without 1 and 10 ppm ZnO NPs of 20 and 200 nm before sowing. When fibers crosslinked with or without 0.15 and 1.5 mg of ZnO NPs of 20 and 200 nm were used, 150 mL of water was supplied in the silica sand, which become 1 and 10 ppm as the final concentration. As control plants, two kinds of control were prepared in this research, which are (i) as control of 1 ppm and 10 ppm ZnO NPs crosslinked with BL, BFSL, and RA, 1 ppm and 10 ppm ZnO NPs solution were used; and (ii) as control for each treatment, BL, BFSL, RA, and solution without ZnO NPs were used. After sowing, four-day-old soybeans were collected and the main-root length, total root-fresh weight, hypocotyl length, and hypocotyl-fresh weight were measured as morphological parameters (Figure 1).

Among BL, BFSL, and RA fibers crosslinked with ZnO NPs, the germination of soybean grown on BFSL and RA fibers was very low, resulting in the morphological parameters not being able to be analyzed (Figure 1). The length of root and hypocotyl of soybean grown in a solution of 10 ppm ZnO NPs with 200 nm increased compared with that of nontreatment; however, their weight did not change (Figure 1). On the other hand, the length of root and hypocotyl of soybean grown on BL-fiber crosslinked with ZnO NPs was promoted at the application of 1 ppm concentration compared with that of nontreatment; additionally, their weight also increased compared with nontreatment (Figure 1). Based on morphological results, BL-fiber crosslinked with 1 ppm ZnO NPs of 200 nm was used for Zn-content measurement and proteomic analysis.

### 2.2. Organ Specificity of Zn Content in Soybean Grown on Fiber Crosslinked with ZnO NPs

To investigate the actions of Zn with organ specificity in soybean, Zn content was measured. Silica sand was treated with 1 ppm ZnO NPs with 200 nm and 1 ppm BL-fiber crosslinked with ZnO NPs with 200 nm. After sowing, root, hypocotyl, and cotyledon were collected from four-day-old soybeans and were measured for Zn content (Figure 2). Soybeans grown on fiber crosslinked with ZnO NPs had higher Zn content in root and hypocotyl than those grown in ZnO NPs solution. Additionally, Zn content in the root was higher than that in the hypocotyl. However, Zn content in cotyledon was the same level between fiber and solution (Figure 2).

### 2.3. Protein Changes in Soybean Grown on Fiber Crosslinked with ZnO NPs

A proteomic analysis with liquid chromatography (LC)-mass spectrometry (MS) analysis was conducted to investigate the cellular mechanism in soybean growth on fiber crosslinked with ZnO NPs. As for the treatments, four kinds of treatments, which are solution/fiber and nontreatment/ZnO NPs treatment, were performed. Proteins extracted from the soybean root, including hypocotyl after treatment, were analyzed. In total, 7948 proteins and 1647 proteins were identified in the protein database of soybean and *A. thaliana*, respectively. The proteomic results of all samples from different groups were compared by principal component analysis (PCA), which showed different accumulation patterns of proteins from different treatments (Figure 3). These results indicated that ZnO NPs largely affected soybean protein contents; additionally, this effect was different between fiber and solution (Figure 3). The relative abundance of proteins from soybean treated with ZnO NPs compared to nontreated soybean (Appendix A), from soybean grown on fiber crosslinked with ZnO NPs compared to nontreated fiber (Appendix A), from soybean grown on fiber compared to a solution (Appendix A), and from soybean grown on fiber crosslinked with ZnO NPs compared to ZnO NPs solution (Appendix A) were listed.

The abundance of 471 proteins differentially changed with fold change >1.5 and <2/3 in the soybean treated with ZnO NPs solution compared with nontreated soybean. Among 471 proteins, 273 proteins increased and 198 proteins decreased in the soybean treated with ZnO NPs solution compared with nontreated soybean (Appendix A). The functional category of identified proteins was obtained using MapMan Bin codes. The increased proteins were categorized in protein metabolism, stress, photosynthesis, transport, and amino acid metabolism in the functional category, while decreased proteins were mainly involved in RNA metabolism, cell wall, and lipid metabolism (Figure 4A).

Furthermore, the abundance of 830 proteins differentially changed with fold change >1.5 and <2/3 in the soybean grown on a ZnO NPs BL-fiber compared with a nontreated BL-fiber. Among 830 proteins, 470 proteins increased and 360 proteins decreased in the soybean grown on a ZnO NPs BL-fiber compared with nontreated BL-fiber (Appendix A). The functional category of identified proteins was obtained using MapMan Bin codes. The increased proteins were categorized in protein metabolism, secondary metabolism, photosynthesis, transport, and amino acid metabolism in the functional category, while decreased proteins were mainly involved in RNA metabolism, stress, signaling, cell construction, and cell wall (Figure 4B).

### 2.4. Immunoblot Analysis of NADPH Oxidoreductase and Tubulins in Soybean Grown on Fiber Crosslinked with ZnO NPs

Immunoblot analysis of NADPH oxidoreductase and alpha/beta tubulins was carried out to better reveal the change of accumulation of proteins from various treatments. Proteins were extracted from the root and hypocotyl of soybeans, which were treated with or without ZnO NPs on the BL-fiber or solution. Extracted proteins (10 µg) were separated on SDS-polyacrylamide gel. The staining pattern of Coomassie brilliant blue was used as a loading control (Appendix A). To confirm changes of significantly accumulated proteins identified by proteomic analysis (Appendix A), the accumulation of NADPH oxidoreductase (Appendix A), alpha-tubulin (Appendix A), and beta-tubulin (Appendix A) was analyzed using immunoblot analysis. The abundance of the NADPH oxidoreductase in root and hypocotyl increased with the treatment of ZnO NPs compared to nontreatment in solution and on fiber (Figure 5A). The abundance of alpha-tubulin in hypocotyl increased with or without the treatment of ZnO NPs on the BL-fiber compared to the solution; however, that in root did not change (Figure 5B). On the other hand, the abundance of beta-tubulin did not show any changes (Figure 5C).

### 2.5. Growth Changes under Flooding Stress of Soybean Grown on Fibers Crosslinked with ZnO NPs

Because ZnO NPs mitigated the risk associated with abiotic and biotic stresses, flooding-stress tolerance was investigated. Soybean seeds were sowed on BL-fiber or in a solution with 1 ppm ZnO NPs with 200 nm. For the non-flooded group, samples were collected at four days after sowing. For the flooded group, soybean seedlings were exposed to two-day-flooding after two-day-germination, and samples were collected. The weight and length of roots including hypocotyl were measured as morphological parameters (Figure 6). Without flooding stress, the weight and length of roots including hypocotyl increased with the treatment of ZnO NPs; however, they did not change under flooding stress, even if they were treated by ZnO NPs (Figure 6).

## 3. Discussions

### 3.1. Fiber Crosslinked with ZnO NPs Improves Soybean Growth through Zn-Content Increase

Biodegradable fibers are useful materials that are widely used in many applications such as biomedicine, filtration, and food packaging [23]. As fabricating fibers, fibers from cellulose, protein, polylactic acid, polyvinyl alcohol, and polycaprolactone were chosen as the focus [24]. Among these, micro/nanofibers expressed superior properties such as mechanical properties, cell proliferation efficiency, and filtration efficiency [25]. However, their effects on plant growth were limited because of chemical fiber spinning. In this study, the effect of three kinds of non-woven fabrics on plant growth was examined.

ZnO NPs increased the plant growth of coffee [10], lettuce [11], maize/cabbage [12], wheat [13], and tobacco [14]. In soybean, ZnO NPs also increased plant growth and seed yield [15,16]. These reports indicated that the application of ZnO NPs improved soybean growth. To develop a more effective method using ZnO NPs for the enhancement of soybean growth, fiber crosslinked with ZnO NPs was produced. In this study, the solution of ZnO NPs with 200 nm promoted soybean growth at the concentration of 10 ppm, while fibers cross-linked with ZnO NPs promoted growth at a 1 ppm concentration (Figure 1). On the other hand, the length of the root and shoot was reduced with ZnO NPs in radish [26] and spinach [27]. Furthermore, a toxic effect on the growth and development of *A. thaliana* was also revealed when the plant was exposed to ZnO NPs [28]. In another study, both positive and negative effects were shown in *Stevia rebaudiana*, depending on the concentration of applied ZnO NPs [29]. These reports with the present study indicate that the response of plants to ZnO NPs depends on the plant species, type, and size/concentration of the NPs.

Zn efficiency and Zn uptake are very important for plant growth and its total content in soil is influenced by several soil properties such as pH, CaCO_3_, organic matter content, and type of crop, as well as cultivars and nutrient interactions in the soil environment [30,31]. A study on the uptake pathways of ZnO NPs in maize root revealed that the majority of the total ZnO NPs undergo dissolution in the exposure medium, and the released Zn ions are only taken up by the root [32]. Only a small fraction of ZnO NPs absorbed on the root surface can cross the root cortex as a result of speedy cell division and root tip elongation, apart from their entry into the vascular system through the gap of the Casparian strip at the sites of the primary-lateral root junction [33]. Based on these reports, ZnO NPs with 20 nm and 200 nm were used in this study.

Soybeans grown on fiber crosslinked with ZnO NPs have higher Zn content in their roots than those grown in ZnO NPs solution (Figure 2). Once NPs enter the plant cells, such as the vascular bundle and stele, they reach the aerial parts [34] and cellular pores [35] through apoplast or symplast pathways and might interact with cellular and sub-cellular organelles [36]. During this penetration and translocation process, they could damage the sub-cellular organization [36]. However, this direct effect of NPs on sub-cellular organelles was caused by small-sized NPs [37,38]. In the present study, because Zn content in soybean was mainly detected in the root (Figure 2), ZnO with 200 nm might not be able to enter the leaf. Furthermore, because Zn content in soybean grown on fiber crosslinked with ZnO NPs at 200 nm was higher than that of soybean grown in solution (Figure 2), ZnO NPs might be effectively held by fiber. However, confirmatory experiments which directly investigate the plant with analytical technique using a microscope will be a challenge in the future.

### 3.2. Fiber Crosslinked with ZnO NPs Enhances Photosynthesis and Secondary Metabolism in Soybeans

Because Zn application results in an appreciable increase in leaf area, the content of chlorophyll, other photosynthetic pigments, and stomatal conductance, plant growth, and seed yield are improved [39,40]. In wheat, the content of photosynthetic pigments, such as chlorophyll *a*, chlorophyll *b*, and total chlorophyll content, was significantly enhanced by the application of ZnO NPs [41]. Additionally, in this study, photosynthesis-related proteins increased with the application of ZnO NPs solution and further increased in soybean grown on fiber crosslinked with ZnO NPs (Figure 4). These results indicate that ZnO NPs positively affect plant growth through photosynthesis in soybeans.

NPs can also increase the content of plant secondary metabolites in the plant. The levels of secondary metabolites such as anthocyanins, chlorophylls [42], and others were reduced in plants treated with excessive concentrations of NPs; and multiple physiological functions such as photosynthesis/transpiration [43], trace element uptake, nitrogen assimilation, and growth were also inhibited [44,45]. On the other hand, the contents of plant-secondary metabolites were significantly increased when the concentrations were suitable [46]. In the present study, proteins categorized in secondary metabolism significantly increased in soybean grown on fiber crosslinked ZnO NPs (Figure 4). This suggests that the concentration of ZnO NPs might become suitable for soybeans when ZnO NPs crosslinked with fiber, resulting in enhanced soybean growth.

### 3.3. NADPH Oxidoreductase Increases in Soybean Treated with ZnO NPs

The abundance of NADPH oxidoreductase, which was identified as one of the significantly accumulated proteins using proteomic analysis (Appendix A), was confirmed using immunoblot analysis. The abundance of NADPH oxidoreductase increased in soybean by ZnO NPs application (Figure 5). Zn is required for the synthesis of tryptophan, which is a precursor of indole-3-acetic acid, and it also has an active role in the production of the essential growth-hormone auxin [47]. NADPH oxidoreductase increased by the application of auxin and was further raised in rice root and callus by the application of Zn [48]. Hypocotyl elongation was shown to be defective in both light and dark conditions in *shy2* and *axr2* mutants [49,50], which were later revealed as gain-of-function mutants of *AUX/IAAs IAA3* and *IAA7*, respectively [51,52]. Additionally, in *A. thaliana*, auxin patterning and cellular growth were linked through a correlated pattern of auxin efflux carrier localization and cortical microtubule orientation [53]. In this study, there was a similar in tubulin (Figure 5), and the length of root and hypocotyl was improved by the application of ZnO NPs (Figure 1). Furthermore, the abundance of NADPH oxidoreductase increased under the same condition (Figure 5). The present results with the previous finding suggest that soybean growth with ZnO NPs might be caused by increasing auxin through NADPH reductase.

### 3.4. ZnO NPs Do Not Enhance Soybean Growth under Flooding Stress but Enhances It without Stress

ZnO NPs mitigated the risk associated with chilling stress in rice [17], salt stress in tomato [18], drought stress in cucumber [54], arsenic stress in soybean [20], and tobacco mosaic virus [21]. In soybean, Ag NPs with 15 nm improved plant growth under flooding stress by increasing the proteins related to amino acid synthesis and wax formation [37]; and Al_2_O_3_ NPs improved plant growth under flooding stress through mitochondrial proteins by regulating membrane permeability and tricarboxylic acid cycle activity [38]. Furthermore, the secondary metabolism of plants has great research value in the quality, flavor, growth/development, and disease therapy [55,56]; and/or tolerance to drought, salt, cold, diseases, and insect pests [57]. These reports indicate that the application of NPs improves stress tolerance; therefore, flooding stress tolerance was investigated for soybean using fiber crosslinked with ZnO NPs (Figure 6). Without flooding stress, soybean growth increased with the treatment of ZnO NPs; unfortunately, however, they did not change under flooding stress, even if they were treated with ZnO NPs (Figure 6). To understand the effect of flooding stress on soybean treated with ZnO NPs, further study is needed to make the process clear.

## 4. Materials and Methods

### 4.1. The Preparation of Fiber Crosslinked with ZnO NPs

To prepare fiber crosslinked with NPs, 0.15 and 1.5 mg of ZnO NPs with 20 and 200 nm (Skyspring Nanomaterials, Huston, TX, USA) were used. They become 1 and 10 ppm ZnO NPs as the final concentration with 150 mL of water. For the fibers, a cupro-long fiber non-woven fabric (BL; Asahi Kasei, Osaka, Japan), a cupro-short fiber spun lace non-woven fabric (BFSL), and a viscose-rayon short fiber spun lace non-woven fabric (RA) were used. Each fiber was dissolved with 2 g/L of an anionic active agent (Score Roll FC-250; Hokko Chemical, Tokyo, Japan) and 1 g/L of sodium carbonate at 90 °C, subjected to scouring treatment/dehydration treatment for 20 min, and dried. Next, 50 g/L of a polyfunctional cationizing agent (Cationone KCN; Lion Specialty Chemicals, Tokyo, Japan) and 7 g/L of sodium hydroxide were dissolved at 90 °C, subjected to cationization treatment/ dehydration treatment for 20 min, and dried. In order to obtain a non-woven fabric crosslinked with 0.15 or 1.5 mg of ZnO NPs, the cationized non-woven fabric was cut into a vertical 15 cm/a horizontal 10 cm and submerged in solution at 90 °C for 20 min, and then subjected to a dehydration treatment to naturally dry. The concentration of ZnO NPs was calculated from the exhaustion rate and the notation, after which the length of the non-woven fabric was cut to adjust the supported amount.

### 4.2. Plant Material

Soybean (*Glycine max* L. cultivar Enrei) was used in this study. Seeds were sterilized in 2% sodium hypochlorite solution and sown on silica sand. Silica sand was treated in 150 mL solution with or without 1 and 10 ppm ZnO NPs with 20 and 200 nm before sowing. When fibers crosslinked with or without 0.15 mg and 1.5 mg of ZnO NPs with 20 and 200 nm were used, 150 mL of water was supplied in the silica sand, which become 1 ppm and 10 ppm as the final concentration. After sowing, seedlings were maintained at 25 °C in a growth chamber illuminated with white fluorescent light (600 μmol m^−2^ s^−1^, 16 h light period/day) and 70% relative humidity. For morphological analysis, 4-day-old soybeans were collected and main-root length, total root-fresh weight, hypocotyl length, and hypocotyl-fresh weight were measured. For proteomic analysis, roots including hypocotyl from 4-day-old soybeans were used. Roots and hypocotyl were collected for the other biochemical assays at 4-day-old soybeans. Three independent experiments were performed as biological triplicates for all experiments. Independent biological replicates were sown on different days.

### 4.3. Protein Extraction

A portion (300 mg) of samples was ground with a mortar and pestle in 500 µL of lysis buffer, which contains 7 M urea, 2 M thiourea, 5% CHAPS, and 2 mM tributylphosphine. The suspension was centrifuged twice with 16,000× *g* at 4 °C for 10 min. The detergents in the supernatant were removed using the Pierce Detergent Removal Spin Column (Pierce Biotechnology, Rockford, IL, USA). The method of Bradford [58] was used to determine the protein concentration and bovine serum albumin was used as the standard.

### 4.4. Protein Enrichment, Reduction, Alkylation, and Digestion

Extracted proteins (100 µg) were adjusted to a final volume of 100 µL, added 400 µL of methanol, and mixed before the addition of 100 µL of chloroform and 300 µL of water. After mixing and centrifugation at 16,000× *g* for 10 min to achieve phase separation, the upper phase was discarded and 300 µL of methanol was added to the lower phase. After centrifugation at 16,000× *g* for 10 min, the pellet was collected as the soluble fraction [59]. Proteins were resuspended in 50 mM ammonium bicarbonate, reduced with 50 mM dithiothreitol for 30 min at 56 °C, and alkylated with 50 mM iodoacetamide for 30 min at 37 °C. Alkylated proteins were digested with trypsin (Wako, Osaka, Japan) at a 1:100 enzyme/protein ratio for 18 h at 37 °C. Peptides were desalted with MonoSpin C18 Column (GL Sciences, Tokyo, Japan) and acidified with 1% trifluoroacetic acid.

### 4.5. Protein Identification Using Nano-Liquid Chromatography Mass Spectrometry

The nano-liquid chromatography (LC) conditions as well as the mass spectrometry (MS) acquisition conditions are described in the previous study [60]. The peptides were loaded onto the LC system (EASY-nLC 1000; Thermo Fisher Scientific, San Jose, CA, USA) equipped with a trap column (Acclaim PepMap 100 C18 LC column, 3 µm, 75 µm ID × 20 mm; Thermo Fisher Scientific), equilibrated with 0.1% formic acid, and eluted with a linear acetonitrile gradient (0–35%) in 0.1% formic acid at a flow rate of 300 nL min^−1^. The eluted peptides were loaded and separated on the column (EASY-Spray C18 LC column, 3 µm, 75 µm ID × 150 mm; Thermo Fisher Scientific) with a spray voltage of 2 kV (Ion Transfer Tube temperature: 275 °C). The peptide ions were detected using MS (Orbitrap Fusion ETD MS; Thermo Fisher Scientific) in the data-dependent acquisition mode with the installed Xcalibur software (version 4.0; Thermo Fisher Scientific). Full-scan mass spectra were acquired in the MS over 375–1500 m/z with a resolution of 120,000. The most intense precursor ions were selected for collision-induced fragmentation in the linear ion trap at a normalized collision energy of 35%. Dynamic exclusion was employed within 60 s to prevent repetitive selection of peptides.

### 4.6. Mass-Spectrometry Data Analysis

The MS/MS searches were carried out using MASCOT (version 2.6.2, Matrix Science, London, UK) and SEQUEST HT search algorithms against the UniProtKB *Glycine max* (Soybean) protein database (version 2021-02; 84,894 proteins) and *A. thaliana* protein database (version 2021-02; 119,669 proteins) using Proteome Discoverer (version 2.4; Thermo Fisher Scientific). The workflow for both algorithms included spectrum files RC, spectrum selector, MASCOT, SEQUEST HT search nodes, percolator, ptmRS, and minor feature detector nodes. Oxidation of methionine was set as a variable modification and carbamidomethylation of cysteine was set as a fixed modification. Mass tolerances in MS and MS/MS were set at 10 ppm and 0.6 Da, respectively. Trypsin was specified as protease and a maximum of two missed cleavage was allowed. Target-decoy database searches were used for the calculation of the false discovery rate, which was set at 1% for peptide identification.

### 4.7. Differential Analysis of Proteins Using Mass Spectrometry Data

Label-free quantification was also performed with Proteome Discoverer using precursor ions quantifier nodes. For differential analysis of the relative abundance of peptides and proteins between samples, the freely available software Perseus (version 1.6.15.0) [61] was used. Proteins and peptides abundances were transferred into the log2 scale. Three biological replicates of each sample were grouped and a minimum of three valid values were required in at least one group. Normalization of the abundances was performed to subtract the median of each sample. Missing values were imputed based on a normal distribution (width = 0.3, down-shift = 1.8). Significance was assessed using Student’s *t*-test analysis. Principal component analysis (PCA) was performed with Proteome Discoverer. Protein functions were categorized using MapMan bin codes [62].

### 4.8. Immunoblot Analysis

SDS-sample buffer consisting of 60 mM Tris-HCl (pH 6.8), 2% SDS, 10% glycerol, and 5% dithiothreitol was added to protein samples [63]. Proteins (10 µg) quantified by Bradford method [58] were separated by electrophoresis on a 10% SDS-polyacrylamide gel and transferred onto a polyvinylidene difluoride membrane using a semidry transfer blotter (Nippon Eido, Tokyo, Japan). The blotted membrane was blocked for 5 min in Bullet Blocking One regent (Nacalai Tesque, Kyoto, Japan). After blocking, the membrane was cross-reacted with a 1: 1000 dilution of the primary antibodies for 1 h at room temperature. As the primary antibodies, the followings were used: alpha-tubulin (Abcam, Cambridge, UK), beta-tubulin (Proteintech, Rosemont, IL, USA), and NADPH-dependent oxidoreductase [48]. Anti-rabbit IgG conjugated with horseradish peroxidase (Bio-Rad, Hercules, CA, USA) was used as the secondary antibody. After 1 h incubation, signals were detected using the TMB Membrane Peroxidase Substrate kit (Seracare, Milford, MA, USA) following the manufacturer’s protocol. Coomassie brilliant blue staining was used as a loading control. The integrated densities of bands were calculated using ImageJ software (version 1.8, National Institutes of Health, Bethesda, MD, USA).

### 4.9. Measurement of Zn Content

A portion (100 mg) of samples, which were root, hypocotyl, and cotyledon, was ground with a mortar and pestle in 250 µL of phosphate-buffered saline containing 0.01 M HCl. The suspension was stirred for 20 min and centrifuged with 16,000× *g* at 4 °C for 10 min. After precipitation of proteins, the supernatant was used for Zn-content assay. Zn content was analyzed using Zn Concentrate Assay Kit (Funakoshi, Tokyo, Japan). Sample (50 µL) was added to 50 µL of chelate color and mixed for 10 min. The absorbance of the mixture was measured at 560 nm.

### 4.10. Statistical Analysis

The statistical significance of the 2 groups was evaluated by the Student’s *t*-test. A *p*-value of less than 0.05 was considered statistically significant.

## 5. Conclusions

To develop a more effective method using NPs for enhancement of soybean growth, fiber crosslinked with ZnO NPs was used. The solution of ZnO NPs with 200 nm promoted soybean growth at the concentration of 10 ppm, while fibers crosslinked with ZnO NPs promoted growth at a 1 ppm concentration. Among three kinds of fibers, BL was the most effective for soybean growth compared to the other fibers. To study the positive mechanism of BL fiber crosslinked with ZnO NPs on soybean growth, a proteomic technique was used. The main findings of this study are as follows: (i) soybeans grown on fiber crosslinked with ZnO NPs had higher Zn content in their root than those grown in ZnO NPs solution; (ii) proteins categorized in photosynthesis and secondary metabolism accumulated more in soybean grown on fiber crosslinked with ZnO NPs than in those grown in ZnO NPs solution; and (iii) the abundance of NADPH oxidoreductase increased in soybean by ZnO NPs application. These findings suggest that BL-fiber crosslinked with ZnO NPs enhances soybean growth through the increase of photosynthesis and secondary metabolism. Furthermore, the accumulation of NADPH oxidoreductase might relate to the effect of auxin with fiber crosslinked with ZnO NPs on soybean growth. Taken together, ZnO NPs at low doses can be considered a potent substance for potential utilization in plant sciences.

## Figures and Tables

**Figure 1 ijms-23-07415-f001:**
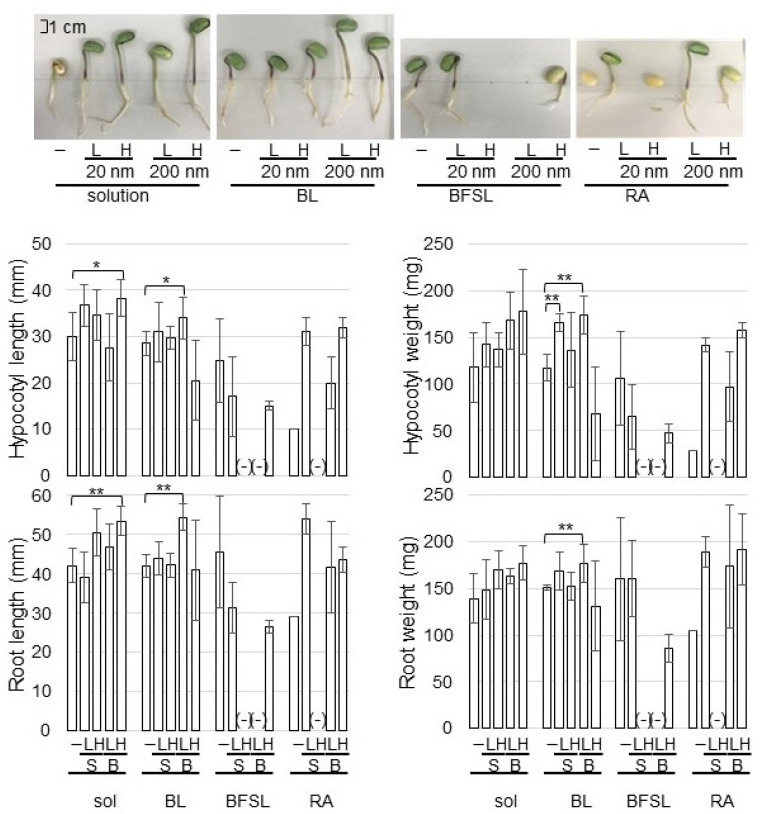
Effects of ZnO NPs on the morphology of soybean seedlings. Silica sand was treated with or without 1 ppm (L) and 10 ppm (H) ZnO NPs with 20 nm (S) and 200 nm (B) before sowing. When fibers of BL, BFSL, and RA crosslinked with or without 0.15 mg and 1.5 mg of ZnO NPs of 20 nm (S) and 200 nm (B) were used, 150 mL of water was supplied in the silica sand, which become 1 ppm (L) and 10 ppm (H) as final concentration. After sowing, four-day-old soybeans were collected and main-root length, total root-fresh weight, hypocotyl length, and hypocotyl-fresh weight were measured as morphological parameters. The scale bar in the photograph indicates 1 cm. “-” means nontreated soybean for each group as control. Data are presented as the mean ± S.D. from three independent biological replicates. Asterisks indicate significant changes between two treatments according to Student’s *t*-test (** *p* < 0.01; * *p* < 0.05).

**Figure 2 ijms-23-07415-f002:**
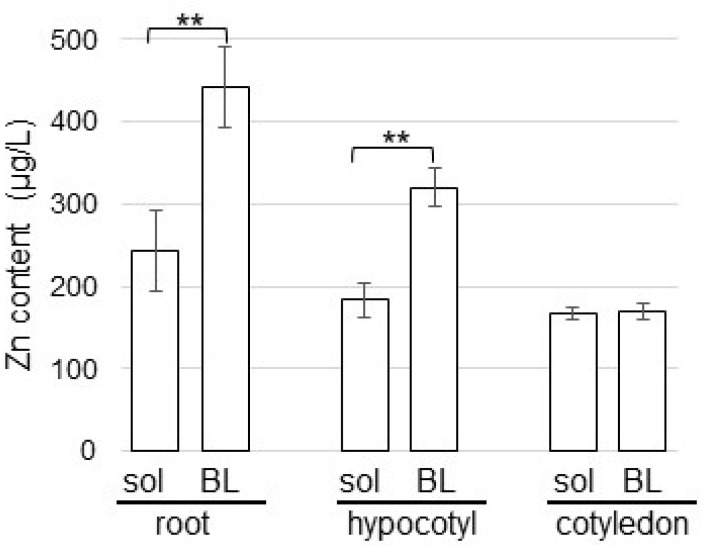
Effects of ZnO NPs on the Zn content in soybean seedlings. Silica sand was treated with 1 ppm ZnO NPs with 200 nm (sol) and 1 ppm BL-fiber crosslinked with ZnO NPs with 200 nm (BL). After sowing, root, hypocotyl, and cotyledon were collected from four-day-old soybeans and the Zn content was measured. Data are presented as the mean ± S.D. from three independent biological replicates. Asterisks indicate significant changes between the two treatments according to Student’s *t*-test (** *p* < 0.01).

**Figure 3 ijms-23-07415-f003:**
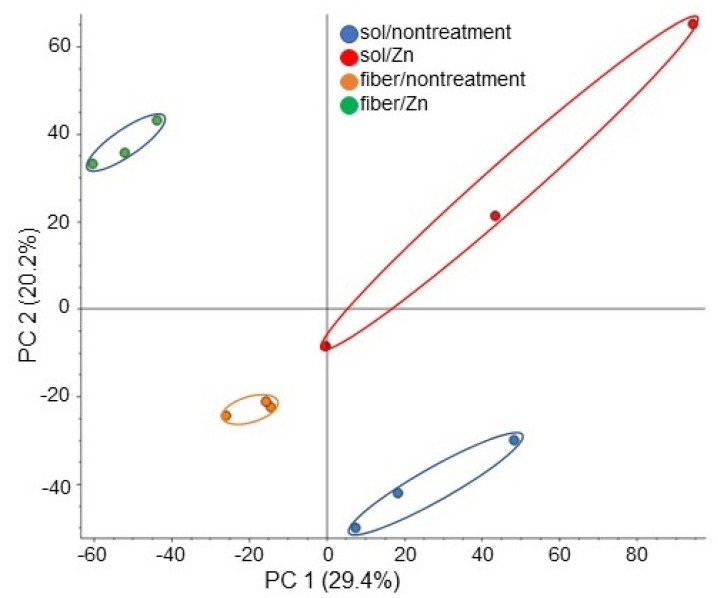
Overview of total proteomic data from 12 samples based on PCA. Proteins from soybean roots including hypocotyl treated with or without 1 ppm ZnO NPs with 200 nm in solution or BL-fiber were extracted and analyzed using a LC-MS. Blue, red, orange, and green colors indicate nontreated solution, ZnO NPs solution, nontreated fiber, and fiber crosslinked with ZnO NPs sample groups, respectively. Three biological replicates were performed in each group.

**Figure 4 ijms-23-07415-f004:**
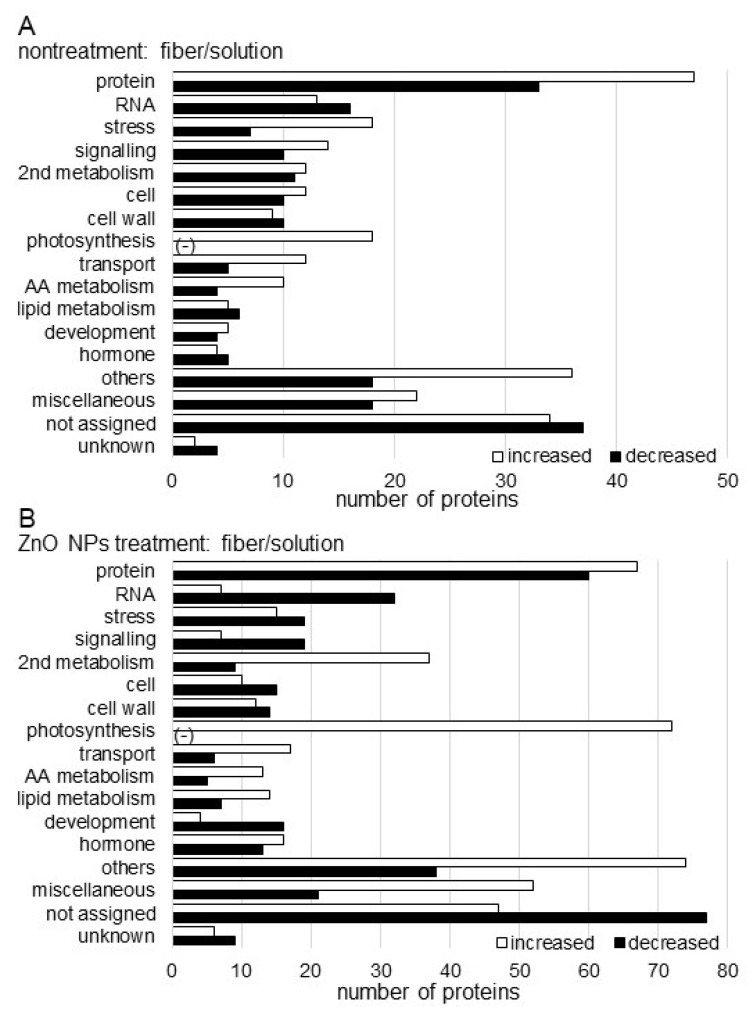
Functional categorization of proteins identified in the root including hypocotyl of soybean treated with ZnO NPs. Proteins extracted from roots including hypocotyl were analyzed using a LC-MS and significantly changed proteins were identified (*p* < 0.05). The significantly changed proteins, 471 and 830 in nontreatment (**A**) and ZnO NPs treatment (**B**), respectively, between solution and BL-fiber, were functionally categorized using MapMan bin codes. The *x*-axis indicates the number of identified proteins. Abbreviations: AA, amino acid; protein, protein synthesis/degradation/post-translational modification/targeting/folding/AA activation; cell, cell division/organization/vesicle transport; RNA, RNA metabolism; hormone, hormone metabolism; 2nd metabolism, secondary metabolism; and others, nitrogen metabolism/metal handling/tetrapyrrole synthesis/oxidative pentose pathway/minor carbohydrate/cofactor and vitamin metabolism/gluconeogenesis. The negative sign shows that there is no identified protein in the respective functional category.

**Figure 5 ijms-23-07415-f005:**
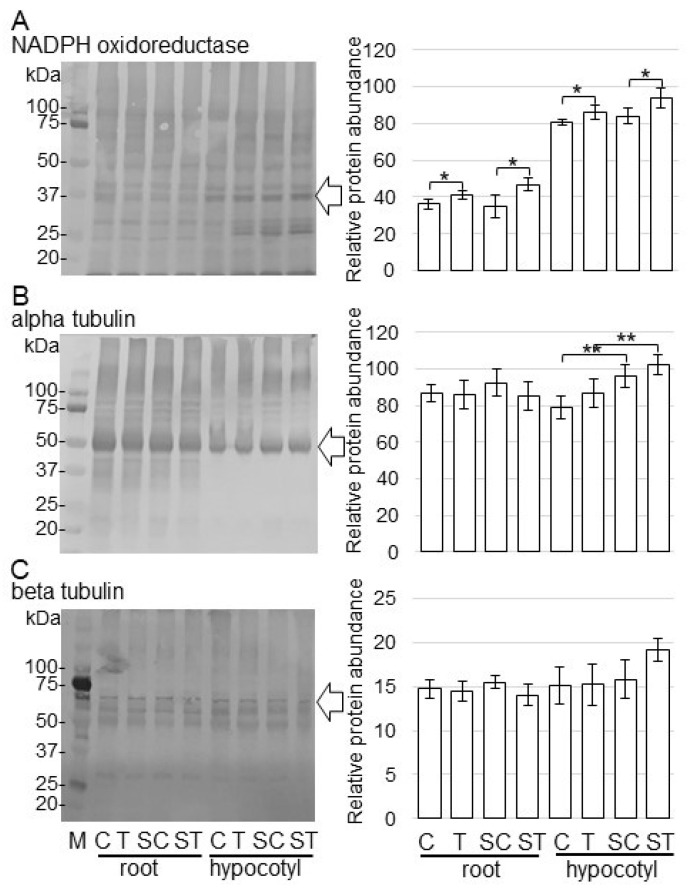
Accumulation of NADPH oxidoreductase and tubulins in root and hypocotyl of soybean treated with ZnO NPs. Proteins (10 µg) extracted from root and hypocotyl of soybean treated with or without ZnO NPs in solution and BL-fiber, separated on SDS-polyacrylamide gel, and transferred onto membranes. The membranes were cross-reacted with anti-NADPH oxidoreductase (**A**), anti-alpha tubulin (**B**), and anti-beta tubulin (**C**) antibodies. Coomassie brilliant blue pattern was used as a loading control (Appendix A). The integrated densities of bands were calculated using ImageJ software. Data are presented as the mean ± S.D. from three independent biological replicates (Appendix A). Asterisks indicate significant changes between the two treatments according to Student’s *t*-test (** *p* < 0.01; * *p* < 0.05).

**Figure 6 ijms-23-07415-f006:**
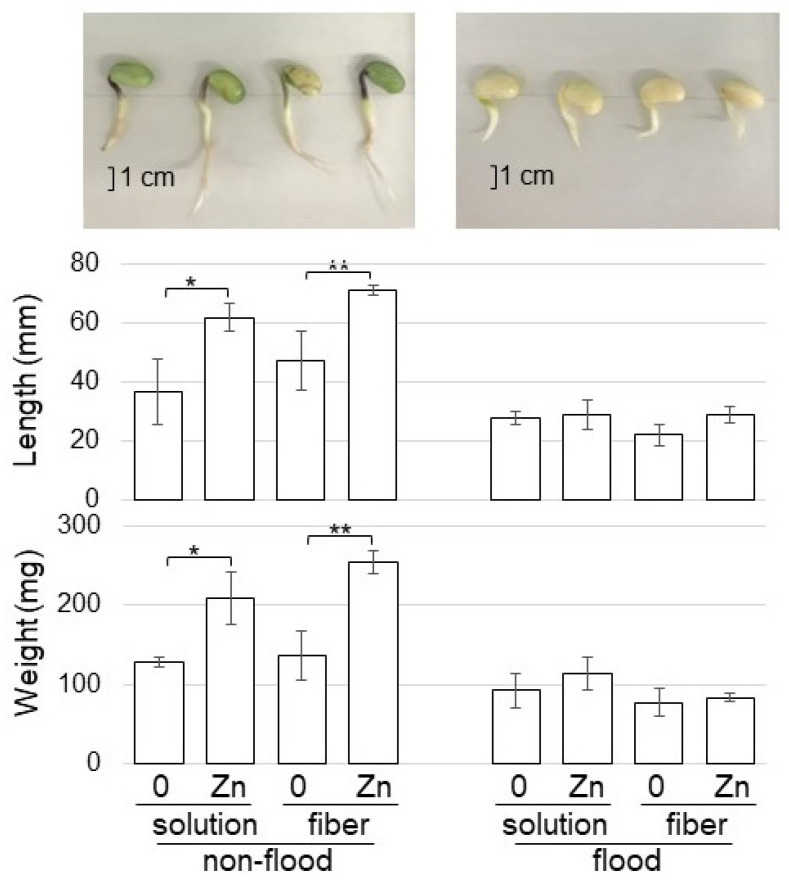
Morphological effect of fiber crosslinked with ZnO on soybean growth under flooding stress. Soybean seeds were sowed on the BL-fiber or solution with 1 ppm ZnO NPs with 200 nm. For the non-flooded group, samples were collected at four days after sowing. For the flooded group, soybean seedlings were exposed to two-day-flooding after two-day-germination, and samples were collected. The fresh weight and length of hypocotyl and root were measured as morphological parameters. The bar in the photograph indicates one cm. The data are given as the mean ± S.D. from three independent biological replicates. Asterisks indicate significant changes between the two treatments according to Student’s *t*-test (** *p* < 0.01; * *p* < 0.05).

## Data Availability

For MS data, RAW data, peak lists, and result files have been deposited in the ProteomeXchange Consortium [64] via the jPOST [65] partner repository under data-set identifiers PXD031670.

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
