# Peer review of "Morphological and Proteomic Analyses of Soybean Seedling Interaction Mechanism Affected by Fiber Crosslinked with Zinc-Oxide Nanoparticles"

_ijms, 2022, doi:10.3390/ijms23137415_

Round 1

Reviewer 1 Report

Extensive changes needed to improve grammar and sentence structure.

Changes in length are NOT reported as ppm.  This makes me think that this is jibberish.  It is repeated throughout the manuscript.

A control is missing from the plant growth experiments.  This control would be applying the same amount of Zn in the solution as was added in the Nano Particles (NP) treatments.  It is not clear if changes in growth were due to Zn leaching from the (NP). If the change is due to Zn ion leaching from the NP, then why use the NP at all. In the growth bar charts, it looks like the solution alone stimulated growth as much as BL(Fig 1). One interpretation of Fig. 1 is that BFSL and RA are just inhibitory to root function and thus there is less hypocotyl growth.

The methods section looks like about right but the grammar is atrocious.

In Fig 4, what was this "gel-free/label-free proteomic technique"? It is repeated in Fig 3.  Why not just say LC-MS?

Fig. 5 should have a control lane with the respective purified protein. If the purified protein could not be acquired then a western blot should be performed showing which band is the actual protein. It is not clear that equal amounts of protein were loaded in each lane.

Author Response

Reviewer 1

Changes in length are NOT reported as ppm.  This makes me think that this is jibberish.  It is repeated throughout the manuscript.

Answer: We are sorry for this problem. The unit of concentration has been corrected and “ppm” has been used in this article. 

A control is missing from the plant growth experiments.  This control would be applying the same amount of Zn in the solution as was added in the Nano Particles (NP) treatments. 

Answer: We are sorry that these sentences were not clear. Two kinds of control were prepared in this research. 1) As control of 1 ppm and 10 ppm ZnO NPs crosslinked with BL, BFSL, and RA, 1 ppm and 10 ppm ZnO NPs solution were used. 2) As control for each treatment, BL, BFSL, RA, and solution without ZnO NPs were used. However, because balk Zn solution did not affect on soybean growth and did not crosslink to fiber, those control could not set up in this research. The results section “2.1” has been corrected in red.

It is not clear if changes in growth were due to Zn leaching from the (NP). If the change is due to Zn ion leaching from the NP, then why use the NP at all. In the growth bar charts, it looks like the solution alone stimulated growth as much as BL (Fig 1). One interpretation of Fig. 1 is that BFSL and RA are just inhibitory to root function and thus there is less hypocotyl growth.

Answer: Thank you very much for your useful comments. Because there is size dependency of nanoparticle for plant growth, we used 20 nm ZnO as the smaller size and 200 nm as the bigger size. The following sentences have been added in the discussion section “3.1”. Corrected parts have been marked in red: “Study on the uptake pathways of ZnO NPs by maize roots revealed that the majority of the total ZnO NPs undergo dissolution in the exposure medium, and the released Zn ions are only taken up by the roots [Lv et al., 2015]. Only a small fraction of ZnO NPs absorbed on the root surface can cross the root cortex as a result of speedy cell division and root tip elongation, apart from their entry into the vascular system through the gap of the Casparian strip at the sites of the primary-lateral root junction [Hossain et al., 2020].”

The methods section looks like about right but the grammar is atrocious.

Answer: We are sorry for this problem. They have been corrected in the methods section “4.1” in red. Especially, the methods section “4.1” has been corrected as follows: “4.1. The Preparation of Fiber Crosslinked with ZnO NPs

To prepare fiber crosslinked with NPs, 0.15 and 1.5 mg of ZnO NPs with 20 and 200 nm (Skyspring Nanomaterials, Huston, TX, USA) were used. They become 1 and 10 ppm ZnO NPs as the final concentration with 150 mL of water. For the fibers, a cupro-long fiber non-woven fabric (BL; Asahi Kasei, Osaka, Japan), a cupro-short fiber spun lace non-woven fabric (BFSL), and a viscose-rayon short fiber spun lace non-woven fabric (RA) were used. Each fiber was dissolved with 2 g/L of an anionic active agent (Score Roll FC-250; Hokko Chemical, Tokyo, Japan) and 1 g/L of sodium carbonate at 90°C, and subjected to scouring treatment/ dehydration treatment for 20 min, and dried. Next, 50 g/L of a polyfunctional cationizing agent (Cationone KCN; Lion Specialty Chemicals, Tokyo, Japan) and 7 g/L of sodium hydroxide were dissolved at 90°C, subjected to cationization treatment/ dehydration treatment for 20 min, and dried. In order to obtain a non-woven fabric crosslinked with 0.15 or 1.5 mg of ZnO NPs, the cationized non-woven fabric was cut into a vertical 15 cm/ a horizontal 10 cm, and submerged in solution at 90°C for 20 min, and then subjected to a dehydration treatment to naturally dry. The concentration of ZnO NPs were calculated from the exhaustion rate and the notation, after which the length of the non-woven fabric was cut to adjust the supported amount.”  

In Fig 4, what was this "gel-free/label-free proteomic technique"? It is repeated in Fig 3.  Why not just say LC-MS?

Answer: As suggested, the term "gel-free/label-free proteomic technique" has been changed to “LC-MS” in this article.

Fig. 5 should have a control lane with the respective purified protein. If the purified protein could not be acquired then a western blot should be performed showing which band is the actual protein. It is not clear that equal amounts of protein were loaded in each lane.

Answer: We are using two methods for application of the equal amounts of protein. Proteins (10 µg) quantified by Bradford method were separated by electrophoresis on a 10% SDS-polyacrylamide gel. Additionally, Coomassie brilliant blue staining was used as a loading control (Figure S2). They have been clearly written in the methods section “4.8” and the results section “2.4” in red.

Reviewer 2 Report

This is a very systematic investigation. I think this paper can be accepted for publication after minor revision.

1. The authors made many speculations about the interaction between ZnO and plant tissues, For example, During this penetration and translocation process, they could damage the sub-cellular organization. However, the author does not appear to have directly investigated his samples with an analytical technique called a living microscope.

2. Why is the sum of the two PCA factors in FIG. 3 less than 50%?

3. Why choose ZnO at 20 and 200 nm?

4. Why the germination of soybean grown on BFSL and RA fibers was very low? What are the rules behind fiber selection?

Author Response

Reviewer 2

This is a very systematic investigation. I think this paper can be accepted for publication after minor revision.

Answer: Thank you very much for your useful comments. Based on your suggestion, this article has been corrected in red.

  1. The authors made many speculations about the interaction between ZnO and plant tissues, For example, During this penetration and translocation process, they could damage the sub-cellular organization. However, the author does not appear to have directly investigated his samples with an analytical technique called a living microscope.

Answer: Thank you very much for your comments. Based on the suggestion, the parts of discussion based on speculation have been reduced or discussed with used previous reports in this article. Corrected parts have been marked in red. For example, the following sentence has been removed from the article: “Therefore, ZnO NPs at low doses might be considered as a potent substance for potential utilization in the field.” Additionally, the following sentence has been added in the discussion section “3.1”: “However, confirmatory experiments which directly investigate the plant with analytical technique using a microscope will be a challenge in the future. “

  1. Why is the sum of the two PCA factors in FIG. 3 less than 50%?

Answer: We appreciate your point. The PCA results mean that there is a large variation in protein expression in treated roots. Although these results do not include all of the proteins detected, they are enough for the inferential purpose of identifying similarities and differences between the samples.

  1. Why choose ZnO at 20 and 200 nm?

Answer: Thank you very much for your suggestion. Because there is size dependency of nanoparticle for plant growth, we used 20 nm ZnO as the smaller size and 200 nm as the bigger size. The following sentences have been added in the discussion section “3.1”. Corrected parts have been marked in red: “Study on the uptake pathways of ZnO NPs by maize roots revealed that the majority of the total ZnO NPs undergo dissolution in the exposure medium, and the released Zn ions are only taken up by the roots [Lv et al., 2015]. Only a small fraction of ZnO NPs absorbed on the root surface can cross the root cortex as a result of speedy cell division and root tip elongation, apart from their entry into the vascular system through the gap of the Casparian strip at the sites of the primary-lateral root junction [Hossain et al., 2020]. Based on these reports, ZnO NPs with 20 nm and 200 nm were used in this study.”

4-1. Why the germination of soybean grown on BFSL and RA fibers was very low?

Answer: We are sorry that it was not identified why the germination of soybean grown on BFSL and RA fibers was very low. However, it was suggested that BL fiber crosslinked with ZnO NPs enhances soybean growth through the increase of photosynthesis and secondary metabolism. Additionally, the accumulation of NADPH oxidoreductase might relate to the effect of auxin with fiber crosslinked with ZnO NPs on soybean growth.

4-2. What are the rules behind selection?

Answer: As suggested, the reason for the selection of 3 kinds of fibers has been added in the discussion section “3.1” as follows: “Biodegradable fibers are useful materials that have been widely used in many applications such as biomedicine, filtration, and food packaging (Deng et al., 2021). As fabricating fibers, fibers from cellulose, protein, polylactic acid, polyvinyl alcohol, and polycaprolactone were chosen as the focus (Soares et al., 2018). Among these, micro/nanofibers expressed superior properties such as mechanical functions, cell proliferation efficiency, filtration efficiency, and antibacterial efficiency (Kai et al., 2014). However, the effect on plant growth was limited because of chemical fiber spinning. In this study, the effect of 3 kinds of non-woven fabrics on plant growth was examined.”

Round 2

Reviewer 1 Report

The manuscript still refers to changes in the length of the hypocotyl in PPM.  This is not the units usually used for changes in length of anything!. See lines 22,23,95,248,471,472.  This must be changed.  The authors state that they changed this in the new version, which they obviously did not. 

It is not clear that these results are translatable to a field.  I would imagine that any increase in yield would be heavily off set by the cost of the cross-linked NP.  If this is true, then these results, while interesting, are not translatable.  This would limit the utility of this manuscript.

If ZnO is anti-bacterial, then it may be detrimental to any plant that relies on endophytes for optimum growth.  This may be involved in the varied results on different plant species.

Author Response

The manuscript still refers to changes in the length of the hypocotyl in PPM.  This is not the units usually used for changes in length of anything!. See lines 22,23,95,248,471,472.  This must be changed.  The authors state that they changed this in the new version, which they obviously did not. 

Answer: Thank you very much for your point. We are sorry that we were mistaken. The sentence has been corrected in blue color. The corrected sentence is as follows: The solution of ZnO NPs with 200 nm promoted soybean growth at the concentration of 10 ppm, while fibers cross-linked with ZnO NPs promoted growth at a 1 ppm concentration.

It is not clear that these results are translatable to a field.  I would imagine that any increase in yield would be heavily off set by the cost of the cross-linked NP.  If this is true, then these results, while interesting, are not translatable.  This would limit the utility of this manuscript.

Answer: Thank you very much for your point. Because the research related to “field work” did not perform in this study, the sentence has been removed as suggested. The corrected parts have been marked in blue in the text.

If ZnO is anti-bacterial, then it may be detrimental to any plant that relies on endophytes for optimum growth.  This may be involved in the varied results on different plant species.

Answer: Thank you very much for your point. Because the research related to “anti-bacterial” did not perform in this study, the words and sentences about “anti-bacterial” have been removed. The corrected parts have been marked in blue in the text.